# Immune Checkpoint Inhibitors (ICI) in Urological Cancers: A New Modern Era, but Not Generally Applied

**DOI:** 10.3390/ijms26157194

**Published:** 2025-07-25

**Authors:** Marcin Sokołowski, Anna Sokołowska, Magdalena Chrząszcz, Aleksandra Butrym

**Affiliations:** 1Dr Alfred Sokołowski Specialist Hospital in Walbrzych, 58-309 Walbrzych, Poland; annabiernacka7@gmail.com (A.S.); lek.mmch@gmail.com (M.C.); aleksandra.butrym@gmail.com (A.B.); 2Lower Silesian Oncology, Pulmonology and Hematology Center, pl. Hirszfelda 12, 53-413 Wroclaw, Poland; 3Department of Cancer Prevention and Therapy, Wroclaw Medical University, ul Borowska 2-13, 50-556 Wroclaw, Poland

**Keywords:** immunotherapy, checkpoint inhibitors, urological cancers

## Abstract

The modern era of systemic treatment of urological cancers is definitely marked by checkpoint inhibitors. Over the past 30 years, checkpoint inhibitors have changed the oncological world, especially in chemoresistant malignancies. Multiple investigations focused on immunotherapy in urological cancers have carved new paradigms and changed clinical guidelines. However, some clinical trials have been blind alleys for systemic therapy. After a scrutinized review of electronic databases, we want to present the natural history and courses of clinical trials in urological malignancies. All of them contribute to expanding the knowledge and experience of clinicians, and some of them improve the prognosis and prolong the overall survival of oncological patients. In conclusion, checkpoint inhibitors open a new modern era in some urological cancers, but not overall. Future perspectives are focused on combination with targeted therapy and could be a new way forward in the systemic treatment of urological cancers.

## 1. Introduction

Urological cancers, such as renal cell cancer (RCC), urothelial cancer (UC), prostate cancer, and germ cell tumors, are global and social challenges in the modern era of systemic therapy. RCC is a disease with a rising worldwide incidence, estimated at 400,000 new cases annually, and a mortality rate approaching 175,000 deaths per year [1]. UC is the tenth most common cancer in the world, with approximately 500,000 new cases and over 210,000 deaths in 2020 [2]. The World Health Organization (WHO) estimates prostate cancer as the third most common diagnosed malignancy, with 1,414,259 cases globally [2]. Germ cell tumors, especially testicular cancer, recorded 74,458 new cases and almost 10,000 deaths worldwide, as estimated by the International Agency for Research of Cancer (IARC) [3]. Systemic therapy—defined as the use of a substance that travels through the bloodstream, reaching and affecting cells—consists of different combinations of chemotherapy, immunotherapy and targeted therapy. In our review, we want to briefly present and summarize the most important clinical trials, focused on immune checkpoint inhibitors (ICI), which have significantly changed the prognosis and natural courses of urological malignancies.

## 2. History and Molecular Mechanism of Checkpoint Inhibitors

The history of ICI began in 1992 with the work of the group of Tasuku Honjo from Kyoto University, who used a subtractive hybridization assay to isolate a cDNA of a new transmembrane protein, which is called a programmed death cell receptor-1 (PD-1) [4]. Initially, researchers could not find a ligand which could activate programmed death cell. However, seven years later, the group of scientists, directed by Honjo, proved that PD-1 is a negative regulator of B-cell response by studying PD-1-deficient mice cell, especially antibody class switching [5]. Subsequently, a group from the Mayo Clinic demonstrated a third member of B7 family, B7-H1, which decreased T-cell proliferation, reduced secretion of IL-2 (interleukin-2) and increased IL-10 secretion. They also excluded CD28, CTLA4 (cytotoxic T-cell antigen 4) or ICOS (inducible T-cell costimulatory) as a potential receptor for this ligand [6]. Freeman and colleagues, in cooperation with the Honjo team, proved that B7-H1 is a first ligand of PD-1 (PD-L1) [7]. At the same time, other investigators indicated a second ligand of the PD-1, known as PD-L2 [8].

Programmed cell death 1 is a member of the Ig superfamily and the CD28/CTLA-4 subfamily, characterized by a single IgV-like domain in the extracellular region, which is composed of a single N-terminal IgV-like domain [4]. It is expressed on CD8+, CD4+ T cells, NK (natural killer) cells, B cells, and tumor-infiltrating lymphocytes (TILs) [9]. It interacts with PD-L1 (programmed death ligand-1), which is expressed on the surfaces of cells such as T cells, B cells, and NK (natural killer) cells. Additionally, double-negative (CD4-, CD8-) thymocytes may also present PD-1 proteins on their surface. Moreover, some research has revealed that PD-1 may play a basic role in selection during TCRβ rearrangement [10]. The presentation of PD-1 on the plasma membrane is induced by the cytokine common γ chain of IL-2, IL-7, IL-15, and IL-21. In memory T-cell colonies, this effect was more demonstrable [11]. The structure of the receptor is divided into three general parts: an about 20-amino-acid stalk is separated by the IgV domain from the plasma membrane, a transmembrane domain, and a cytoplasmic tail containing two tyrosine-based signaling motifs. The N-terminal extremity sequence contains an immunoreceptor tyrosine-based inhibitory motif (ITIM), called VDYGEL [12], which is required for recruiting SH2 domain-containing phosphatases. The ITIM is composed of a sequence of amino acids located intracellulary in cytoplasmatic domains. The sequence are similar: Ser/Iso/Val/Leu/x/Tyr/x/x/Iso/Val (where x represents any amino acid, Tyr–tyrosine, Ser–serine, I–Isoleucine, V–valine, and Leu–leucine). The tyrosine in the ITIM structure can be phosphorylated, enabling the recruitment of SH2 domain-containing phosphatases, which inhibit cellular activation [13].

The C-terminal extremity contains the sequence TEYATI, an immunoreceptor tyrosine-based switch motif (ITSM), important for the inhibitory function of PD-1 [14]. The ligand is bound to the extracellular region of the receptor. Afterwards, the tyrosine molecules in the ITIM and ITSM sequences, located in the cytoplasmic domain, are dephosphorylated by Src-family tyrosine kinase. Recruited SHP-1 and SHP-2 are used in pathways like RAS/MEK/ERK or PI3K/Akt to stop the cell cycle [15]. SHPs may additionally block the protein kinase c-θ and ZAP70 [16].

The PD-L1, also called protein B7-H1 or CD274 PD-1, is encoded on chromosome 9p24.1 on the CD274 gene [6]. Physiologically, it is expressed on the surface of T cells, B cells, macrophages, dendritic cells, mesenchymal stem cells, myeloid cells and bone marrow-derived mast cells [9].

PD-L2, known as CD273 or B7-DC, is encoded as PDCD1LG2 on chromosome 9p24.1 [8]. It is expressed on an activated CD4+ or CD8+ subset, activated dendritic cells, macrophages, bone marrow-derived mast cells, and more than 50% of peritoneal B1 cells [12,17].

The ligation of PD-L1 to PD-1 leads to the formation of a PD-1/TCR inhibitory microcluster, recruits SHP1/2 and dephosphorylates multiple members of the TCR signaling pathway. This leads to the shut-off of T-cell activation through the induction of apoptosis, reduction in proliferation, and inhibition of cytokine secretion. The entire programmed death pathway is crucial for immune regulation to maintain the balance in T-cell activation and cell tolerance [18]. PD-L1 also transmits anti-apoptotic signals to cancer cells to protect them from apoptosis. In some neoplasms, for example colon cancer, the role of PD-L1 in carcinogenesis is proved [19]. PD-L1 not only inhibits T-cell proliferation and cytokine production but also enhances T-cell activation [20]. The activation of T cells may be induced in several mechanisms, for example the PI3K-AKT pathway, and by cytokines, especially IFN-γ [21]. The cytokines induce expression of PD-L1 in tumor cells as well as in dendritic cells and macrophages [22]. T cells, through PD-L1 expression, initiate an immunosuppressive pathway after recognizing tumor antigens and producing tumor-specific responses (cytokines), by selective inhibition of the cytotoxic T-cell function. The potential PD-1 blockade (ICI) induces the expansion of T cells in a tumor microenvironment [23]. The process of PD-1 ligation to its ligand is presented in Figure 1.

Multiple PD-1 inhibitors, such as nivolumab, pembrolizumab, dostarlimab, cemiplimab, and PD-L1 inhibitors like atezolizumab, avelumab, and durvalumab, play an important role in modern therapy for malignancies, known as immune checkpoint inhibitors [24].

## 3. Predictive Biomarkers for Checkpoint Inhibitor Therapy

In recent years, checkpoint inhibitors have been generally applicable, which has led to the isolation of several predictive biomarkers. The first is mismatch repair deficiency (dMMR), which results from the loss of one or more MMR (mismatch repair) proteins: MLH1, MSH2, MSH6 or PMS2. This deficiency impairs the system responsible for recognizing and repairing erroneous insertion, deletion, and mis-incorporation that occur during DNA replication and recombination. The inability to correct these errors during replication leads to the accumulation of DNA errors in the cell genome and the formation of novel microsatellite fragments, which is called microsatellite instability (MSI) [25]. The MSI can be divided into high (MSI-H), when instability is detected in 30% or more of the tested markers, and low (MSI-L), with a detection >0% but <30% of markers. The microsatellite stability (MSS) is characterized by 0% detection [26]. The dMMR is associated with the Lynch syndrome and benefits from checkpoint inhibitor therapy [27,28]. Chandran and colleagues conducted a review of the prevalence of dMMR and MSI-H in UC. They estimated the prevalence of dMMR in bladder cancer at 2.3% (95% CI 1.12% to 4.65%) and in the upper urinary tract at 8.95% (95% CI 6.81% to 11.67%). The MSI-H were evaluated at 2.11% (95% CI 0.82% to 5.31%) and 8.36% (95% CI 5.50% to 12.53%), respectively [29]. Another team calculated the prevalence of MSI-H/dMMR in prostate cancer at 3.1% [30]. In testicular cancer, the overall prevalence of MSI-H is probably between 1.84% and 4.50% [31]. Tumors with dMMR or/and MSI-H are associated with a higher expression of tumor neoantigens, which simplify immune recognition and increase the chances of a response to ICI therapy [25]. The connection dMMR/MSI-H with ORR (50–90%) was proved by several investigations focused on non-colorectal solid tumors [32,33] and, in small probes, on upper urothelial tract cancer [34].

The tumor mutational burden (TMB) is defined as the number of somatic mutations per megabase (MB) of interrogated genomic sequence. A high TMB, which is commonly defined as more than 20 mutations per MB, is connected with increased production and presentation of neoantigens to cytotoxic cells activated by exposure to immune checkpoint inhibitors. In the literature, intermediate (>5 and <20), low (≤5) and very high TMBs (≥50) have also been described as possible therapy predictors [35]. The prevalence of a high TMB is estimated at 26% in one study, but the definition assumed more than 10 mutations/MB [36]; in another, prevalence was definitely lower, at 8.2% [37]. In a large meta-analysis of 2499 cases of urothelial cancers, the patients with a high TMB reached a significantly longer OS and PFS than those with a low TMB after ICI treatment (OS: HR 0.69, 95% CI 0.62, 0.76, *p* < 0.05; PFS: HR 0.67, 95% CI 0.59, 0.76, *p* < 0.05). The response to ICI therapy was superior in the TMB-high group compared to the TMB-low cohort, but without a significant difference (OR 1.64, 95% CI 0.94, 2.86, *p* = 0.08). The above meta-analysis proved a role of TMB as a predictive biomarker for ICI therapy in UC [38].

The last predictors include two simple methods for measuring PD-L1 expression on the surfaces of tumors cells: TPS (Tumor Proportion Score) and CPS (Combined Positive Score). The TPS is defined as the number of positive tumor cells divided by the total number of viable tumor cells multiplied by 100%. CPS is the number of positive tumor cells, lymphocytes and macrophages, divided by the total number of viable tumor cells multiplied by 100 [39]. The cutoff values for positive TPS or CPS are variable and depend on histopathology and type of neoplasm. The scores have practical significance in non-small lung cancer [40], gastric cancer [41], squamous cell head and neck cancer [39] and cervical cancer [42]. Moreover, the role of the above scores in urological UC are limited to adjuvant therapy for muscle-invasive urothelial cancer with nivolumab after radical surgery (TPS ≥ 1%) [43].

## 4. Renal Cell Cancer (RCC)—Evolution to Revolution

The experience with checkpoint inhibitors in renal cancer (RCC) has been clinically proven and has remained the standard of care for several years. Initial research focused on nivolumab as a second-line treatment for metastatic renal cell carcinoma. A phase III clinical trial, CheckMate 025, compared nivolumab to everolimus in subsequent-line systemic therapy. In a population of 821 patients with locally advanced or metastatic RCC, the overall response rate (ORR) was higher in the nivolumab cohort (25% vs. 5%). Furthermore, immunotherapy statistically improved overall survival (OS) from 19.6 months to 25 months, regardless of PD-L1 expression. The quality of life was better in the nivolumab population. In November 2015, the FDA approved the agent, after progression following one or two regimens of antiangiogenic therapy [44]. The real-life population study NORA confirmed the effectiveness and safety of nivolumab in the second-line therapy for RCC [45].

The next step was to transfer checkpoint inhibitors to the first line of RCC treatment. The natural course led to an anti-CTLA4/anti-PD1 inhibitor doublet. The phase III clinical trial, CheckMate 214, compared the combination of nivolumab and ipilimumab to the standard of care, sunitinib, in a population of 1096 untreated patients with advanced or metastatic RCC, with poor or intermediate disease risk. The ORR was statistically higher in the doublet group (42% versus 27%, respectively), and PFS (progression-free survival) improved in the treatment arm to 22.8 months vs. (versus) 5.9 months (the group with >1% PD-L1 expression). It was noted that in the unfavorable risk group, outcomes were better in the sunitinib group [46]. In the first publication in 2017, OS was not reached in the experimental arm; however, the last publication from ASCO (American Society of Clinical Oncology) in 2024 showed long-term survival (76 months vs. 40 months for long responders) and a longer duration of response (NIVO + IPI: *n* = 137 vs. SUN: *n* = 80 after 6 years). Complete response (CR) rates were also statistically higher (27% vs. 9%) [47]. The FDA approved nivolumab with ipilimumab in previously untreated advanced RCC with poor or intermediate risk.

Further trials focused on combining checkpoint inhibitors with an antiangiogenic agent (VEGFR). In a phase II randomized clinical trial, IMmotion 150, involving advanced or mRCC, the combination of atezolizumab with bevacizumab improved PFS to 6.1 months compared to either agent alone. However, the combination was not better than the control arm, sunitinib [48]. The next direction was the investigation with multifocal tyrosine kinase inhibitors. The first was KEYNOTE-426, which compared pembrolizumab with axitinib versus sunitinib in a population of 861 untreated patients with advanced RCC. After 12 months, the OS improved in the doublet arm (90% vs. 78%), as did median PFS (15.1 months vs. 11.1 months) [49]. After 5 years of follow-up, pembrolizumab + axitinib sustained OS (41.9% vs. 37.1%), PFS (18.3% vs. 7.3%) and ORR benefits over sunitinib in advanced ccRCC [49]. Subsequently, a JAVELIN Renal 101 trial compared avelumab with axitinib to sunitinib in a population of 821 patients with untreated advanced RCC. PFS improved in the experimental arm (13.8 vs. 8.4 months) [50]. Unfortunately, after 5 years of follow-up, analyses favored OS for avelumab + axitinib vs. sunitinib but did not reach statistical significance; that PFS was longer (13.9 months vs. 8.5 months) [50] seems to be the result of the clinical trial CheckMate 9ER, which compared the combination of nivolumab with cabozantinib to sunitinib in a population of previously untreated 651 patients with advanced renal cell carcinoma. In the experimental arm, PFS improved to 16.6 months vs. 8.3 months in the sunitinib group (*p* < 0.001). Furthermore, ORR was 55.7% in the experimental arm vs. 27.1% in patients receiving sunitinib (*p* < 0.001). Adverse events of any cause of grade 3 occurred similarly (75.3% vs. 70.6%). Survival outcomes favored nivolumab with cabozantinib over sunitinib across intermediate, poor and combined intermediate/poor IMDC (International Metastatic Renal Cell Carcinoma) risk subgroups [51]. The evolution of clinical trials has led to a revolution in systemic treatment of metastatic/advanced RCC, and a combination of nivolumab and cabozantinib is the most preferred option for the first-line therapy of the NCCN (National Comprehensive Cancer Network) [52] and ESMO (European Society for Medical Oncology) [53]. A similar combination—pembrolizumab and levantinib—was also studied in a phase III CLEAR trial with three arms: lenvatinib plus pembrolzumab, levantinib plus everolimus, and sunitinib monotherapy. Progression-free survival was longer in the lenvatinib plus pembrolizumab arm compared to sunitinib (median, 23.9 vs. 9.2 months; hazard ratio for disease progression or death, 0.39; 95% confidence interval [CI], 0.32 to 0.49; *p* < 0.001). Furthermore, the levantinib plus everolimus arm had also a longer PFS than sunitinib (median, 14.7 vs. 9.2 months; hazard ratio, 0.65; 95% CI, 0.53 to 0.80; *p* < 0.001). Overall survival was longer with lenvatinib plus pembrolizumab than with sunitinib (hazard ratio for death, 0.66; 95% CI, 0.49 to 0.88; *p* = 0.005) but was not longer with lenvatinib plus everolimus than with sunitinib (hazard ratio, 1.15; 95% CI, 0.88 to 1.50; *p* = 0.30). Adverse events of any grade were noticed in 82.4% of the patients who received lenvatinib plus pembrolizumab, 83.1% of those who received lenvatinib plus everolimus, and 71.8% of those who received sunitinib. Serious adverse events (grade 3 or higher) occurring in at least 10% of the patients in any group included hypertension, diarrhea, and elevated lipase levels [54].

Based on the new horizon of therapy in metastatic/advanced RCC, immunotherapy is useful in adjuvant treatment. After a string of failures of TKI such as sorafenib [55], pazopanib [56], and axitinib [57], a qualified success belongs to sunitinib in a S-TRAC trial. After 1 year of adjuvant therapy in a population of 615 patients with high-risk recurrent RCC after nephrectomy, median DFS (Disease-Free Survival) improved to 6.8 months vs. 5.6 months in placebo group (*p* = 0.03), although median OS was not reached by sunitinib [58].

Besides the fact that sunitinib was approved by FDA in November 2017, the therapy did not meet the expectations of clinicians and mobilized intensive trials of immunotherapy. In the randomized phase III trial KEYNOTE-564, one year of adjuvant therapy with pembrolizumab (17 cycles at a dose of 200 mg) was compared to placebo in intermediate (pT2-3 N0) or high-risk clear-cell renal cell carcinoma (pT4 or N+ or M1 after metastasectomy or radiotherapy). After 30 months, the DFS was met with an HR of 0.63 (95% CI 0.50–0.80, *p* < 0.0001), and OS was significantly longer in the pembrolizumab arm (HR 0.52, 95% CI 0.31–0.86, *p* = 0.0048) [59]. Finally, in 2021, FDA approved pembrolizumab as the first valuable agent in adjuvant therapy for RCC [60]. A similar trial, IMmotion010, based on one year of therapy with atezolizumab (1200 mg every 3 weeks for 16 cycles), was unsuccessful. Median investigator-assessed disease-free survival in the experimental arm was not significantly superior to the placebo arm (57.2 months vs. 49.5 months; hazard ratio 0.93, 95% CI 0.75–1.15, *p* = 0·50) [61]. Interestingly, the ICI doublet of nivolumab/ipilimumab in a double-blind, randomized, phase III CheckMate 914 trial was negative (nivolumab 240 mg every 2 weeks + ipilimumab 1 mg/kg). The median disease-free survival was not better in nivolumab + ipilimumab versus the placebo group (*p* = 0·53) [62]. Noteworthily, perioperative nivolumab before nephrectomy, followed by adjuvant nivolumab, did not improve recurrence-free survival versus surgery only, followed by surveillance, in patients with high-risk renal cell carcinoma (33% vs. 33% had recurrence-free survival events, *p* = 0.32), as proved by the PROSPER trial [63].

In conclusion, as of now, only pembrolizumab is an approved and effective ICI in adjuvant treatment after radical surgical/radiotherapeutic curation. We present the positive phase III clinical trials focused on ICI in therapy for RCC in Table 1.

## 5. Urothelial Cell Carcinoma (UCC): Multiple Ways with Many Happy Endings

The first reports from clinical trials focused on immunotherapy in metastatic/advanced UCC were unsatisfactory for researchers and clinicians. That fact was probably connected with three independent ways of investigation. From these directions, the first looks like a small country path, the second like a blind alley, and the third pretends to be a highway to prolongation of PFS.

The first one was therapy with checkpoint inhibitors prior to platinum-based chemotherapy. This approach was especially dedicated to patients who were ineligible for any platinum therapy. In an IMvigor210 phase II trial, atezolizumab was studied in a cohort of patients with metastatic UCC who were ineligible for platinum therapy. The objective response rate was 23% (95% CI 16–31), median PFS was 2.7 months, and median OS was 15.9 months [64]. Further investigation with pembrolizumab in the KEYNOTE-052 phase II single-arm study evaluated the efficacy and safety of the first-line pembrolizumab for patients with locally advanced or metastatic cisplatin-ineligible UCC. Similar to IMvigor210, ORR was 28.6% (95% CI, 24.1% to 33.5%), median OS was 11.3 months (95% CI, 9.7 to 13.1 months), and 12- and 24-month OS rates were 46.9% and 31%. However, in a population with CPS ≥ 10 (combined proportion score), ORR (47.3%) and median OS (18.5 months) were significantly better. Also, in the group of patients with lymph node-only disease, ORR was 49.0% (95% CI, 34.8% to 63.4%), and median OS was 27.0 months (12.4 months to NR). The conclusion was that pembrolizumab is an option in cisplatin-ineligible patients with advanced UCC and is associated with prolonged OS, particularly with PD-L1 CPS ≥ 10 and lymph node-only disease [65]. The FDA ultimately decided to approve therapy with pembrolizumab as first-line systemic therapy in the population of patients ineligible for any chemotherapy [66]. In conclusion, that way of investigation was similar to a small path for a small population of patients, with suboptimal therapy.

The second approach focuses on trials with checkpoint inhibitors after progression following platinum-based chemotherapy. In the phase II trial IMvigor210, patients with UCC who had progressed after chemotherapy were treated with atezolizumab and compared to historical controls of chemotherapy regimens. The ORR improved to 15% from 10% in the historical group. Better response was related to higher PD-L1 expression [67]. The next trial, IMvigor211, was a phase III randomized controlled trial divided into two cohorts of patients with advanced/metastatic UCC, who received either atezolizumab (1200 mg) or chemotherapy (physician’s choice: vinflunine at 320 mg/m^2^, paclitaxel at 175 mg/m^2^, or 75 mg/m^2^ docetaxel). Atezolizumab was not associated with significantly longer overall survival than chemotherapy (11 months vs. 10.6 months), and ORR was similar (23% vs. 22%). However, the safety profile for atezolizumab was favorable compared with chemotherapy (intention-to-treat population grade 3–4 treatment-related adverse events 20% vs. 43%) [68]. In the phase 1b dose-expansion JAVELIN Solid Tumor trial, avelumab was tested in the population of patients after progression following previous platinum chemotherapy. The ORR was 16.5% (95% CI 11.9–22.4%) [69]. Based on this, the FDA approved avelumab in 2017 in treatment of advanced UCC after progression on platinum-based chemotherapy, without phase III trial data [70]. Pembrolizumab showed activity in the multi-cohort, open-label, phase 1b basket trial KEYNOTE-012. Patients received 10 mg/kg intravenous pembrolizumab every 2 weeks until disease progression, unacceptable toxic effects, or the end of the study (24 months). The ORR was achieved in 26% of patients [71]. In the KEYNOTE 045 study, patients with UCC after platinum-based chemotherapy progression received pembrolizumab (200 mg) every 3 weeks or chemotherapy (paclitaxel, docetaxel, or vinflunine). The median OS was higher in the pembrolizumab arm compared to chemotherapy (mOS 10.3 vs. 7.4 months; *p* = 0.002), and the ORR was significantly higher for pembrolizumab (21.1% vs. 11.4%, *p* = 0.001), with no difference in median PFS. Furthermore, the pembrolizumab group had lower rates of treatment-related adverse events (60.9% vs. 90.2%) [72]. The phase II, single-arm CheckMate 275 trial focused on nivolumab in a population of patients with metastatic or surgically unresectable UCC who had previously received platinum-based chemotherapy. The ORR was 19.6% (95% CI 15.0–24.9%). The treatment-related adverse events in grade 3–4 were noted in 18% of participants [73]. Similar to avelumab, the FDA approved nivolumab in 2017 for the treatment of advanced UCC after progression on platinum-based chemotherapy, without phase III trial data [74]. To conclude the second approach, we have to accentuate the unclear role for any single-agent ICI in the second line of therapy for advanced/metastatic UCC.

The new way of therapy is consolidation immunotherapy after response to platinum-based chemotherapy. In the JAVELIN Bladder 100 trial, among 700 patients with unresectable locally advanced or metastatic urothelial cancer who did not have disease progression with first-line chemotherapy, the cohort was divided into an experimental arm—maintenance with avelumab—and the control—best supportive care. The primary endpoint was OS, secondary PFS and safety. The prolongation of OS was significant compared to the control arm (median 21.4 months vs. 14.3 months; hazard ratio [HR] for death, 0.69; 95% confidence interval [CI], 0.56 to 0.86; *p* = 0.001). Furthermore, the median progression-free survival was also longer in the avelumab group (3.7 months vs. 2.0 months; hazard ratio for disease progression or death, 0.62; 95% CI, 0.52 to 0.75), especially in a PD-L1-positive population (5.7 months and 2.1 months; hazard ratio, 0.56; 95% CI, 0.43 to 0.73). The incidence of adverse events from any cause was 98.0% in the avelumab group and 77.7% in the control group; the incidence of adverse events of grade 3 or higher was 47.4% and 25.2%, respectively [75]. In 2017, the FDA approved avelumab in maintenance therapy for urothelial cancer [76]. The latest reports suggest that avelumab could change a natural course of urothelial cancer, which has been proved by real-world data from the AVENANCE study [77] and the Italian cohort READY [78].

A novel possibility is also a combination with a platinum regimen from the first course of therapy, which is being tested in a phase III, multinational, open-label trial, Checkmate 901. The participants with previously untreated unresectable or metastatic urothelial carcinoma were cohorted into two arms. In the first one, they received a combination of nivolumab (at a dose of 360 mg) plus gemcitabine–cisplatin every 3 weeks for up to six cycles, followed by nivolumab (at a dose of 480 mg) every 4 weeks for a maximum of 2 years. In the control arm, the patients received only chemotherapy gemcitabine–cisplatin every 3 weeks for up to six cycles. The overall survival was significantly longer in the nivolumab arm than chemotherapy alone (hazard ratio for death, 0.78; 95% confidence interval [CI], 0.63 to 0.96; *p* = 0.02); the median survival was 21.7 months (95% CI, 18.6 to 26.4) as compared with 18.9 months (95% CI, 14.7 to 22.4), respectively. The PFS was also longer in the nivolumab arm (hazard ratio for progression or death, 0.72; 95% CI, 0.59 to 0.88; *p* = 0.001). The median progression-free survival was 7.9 months and 7.6 months, respectively. The ORR was 57.6% with nivolumab combination therapy and 43.1%, respectively [79].

Nivolumab also found a role in adjuvant therapy after surgery in muscle-invasive urothelial carcinoma. In a phase III, multicenter, double-blind, randomized, controlled trial, CheckMate 274, participants who had undergone radical surgery were cohorted into two arms: nivolumab (240 mg intravenously) or placebo every 2 weeks for up to 1 year. Neoadjuvant cisplatin-based chemotherapy before trial entry was allowed. The median DFS in the intention-to-treat population was longer in the nivolumab arm (20.8 months vs. 10.8 months). The percentage of patients who were alive and disease-free at 6 months was 74.9% with nivolumab and 60.3% with placebo (hazard ratio for disease recurrence or death, 0.70; 98.22% CI, 0.55 to 0.90; *p* < 0.001), especially in the population with expression of 1% or more (74.5% and 55.7%, respectively; hazard ratio, 0.55; 98.72% CI, 0.35 to 0.85; *p* < 0.001). The median survival free from recurrence outside the urothelial tract was 22.9 months (95% CI, 19.2 to 33.4) in the nivolumab arm compared to 13.7 months (95% CI, 8.4 to 20.3) in the placebo arm [43].

In 2024, the combination of ICI with enfortumab vedotin created a new path (probably the fourth) in the treatment of urothelial cancer. The trial randomized 886 patients, who received a combination of pembrolizumab and enfortumab vedotin (*n* = 442) or platinum-based chemotherapy (*n* = 444). As a result, progression-free survival was longer in the enfortumab vedotin—pembrolizumab group than in the chemotherapy group (median, 12.5 months vs. 6.3 months; HR for disease progression or death, 0.45; 95% CI, 0.38 to 0.54; *p* < 0.001). Furthermore, the overall survival was also longer in the experimental arm (median, 31.5 months vs. 16.1 months; HR for death, 0.47; 95% CI, 0.38 to 0.58; *p* < 0.001). Treatment with the combination was also safer than chemotherapy (treatment-related adverse events of grade 3 or higher: 55.9% vs. 69.5%) [80]. As with other paths, the FDA approved a combination of enfortumab vedotin with pembrolizumab in therapy for advanced/metastatic urothelial cancer [81]. We present the positive clinical trials focused on ICI in therapy for UCC in Table 2.

## 6. Prostate Cancer—Attempt to Be Booster

Therapy with ICI in prostate cancer is strongly connected with the occurrence of microsatellite instability-high (MSI-H), a deficient mismatch repair (dMMR) gene, or tumor mutational burden (TMB). The MMR/MSI-H is reported in only 2.2–12% cases of prostate cancer [30,82,83]. In that narrow population, ICI could pretend to be a booster of standard therapy with docetaxel or next-generation hormonal agent/therapy (NHA/NHT), which is being tested in multiple clinical trials in metastatic castration-resistant prostate cancer (mCRPC) and hormone-sensitive prostate cancer (mHSPC).

The KEYNOTE-921 trial included 1030 patients with metastatic castration-resistant prostate cancer (mCRPC), who were randomized 1:1 to receive 200 mg pembrolizumab every 3 weeks or placebo for ≤35 cycles, in combination with 75 mg/m^2^ docetaxel every 3 weeks for ≤10 cycles and 5 mg prednisone. The trial was negative; the dual primary endpoints of median radiographic progression-free survival (rPFS) were similar in both arms (8.6 months for pembrolizumab + docetaxel vs. 8.3 months with placebo + docetaxel; HR 0.85, 95% CI 0.71–1.01; *p* = 0.0335). Furthermore, median OS was not met (19.6 months vs. 19.0 months.; HR 0.92, 95% CI 0.78–1.09; *p* = 0.1677) [84]. Similarly, the KEYNOTE 641 trial, based on comparison of pembrolizumab + enzalutamide vs. placebo + enzalutamide in a population of chemo-naïve mCRPC with or without prior abiraterone therapy, was negative. The median of rPFS (10.4 months with vs. 9.0 months HR 0.98, 95% CI 0.84–1.14; *p* = 0.41) and the median of OS (24.7 months vs. 27.3 months; HR 1.04, 95% CI 0.88–1.22; *p* = 0.66) were not met [85]. The combination of pembrolizumab with olaparib versus NHA in a population of patients who progressed on or after abiraterone/enzalutamide and docetaxel did not show superiority. The median rPFS was similar (4.4 months vs. 4.2 months; HR, 1.0, 95% CI, 0.82 to 1.25; *p* =0.55) as was the median of OS (15.8 months vs. 14.6 months; HR, 0.94 [95% CI, 0.77 to 1.14; *p* = 0.26). Although the ORR was higher with the pembrolizumab group (16.8% vs. 5.9%), the toxicity of therapy was higher for that arm (grade 3: 34.6% vs. 9.0%) [86]. As we can conclude, pembrolizumab is not an optimal booster for standard therapy for mCRPC.

The clinical trials with pembrolizumab also focused on populations with mHSPC. In a double-blind KEYNOTE-991 trial, patients were randomized 1:1 to receive pembrolizumab (200 mg) every 3 weeks or placebo with enzalutamide (160 mg) orally daily + continuous ADT (androgen deprivation therapy). The rPFS in both arms was not reached (HR 1.20, 95% CI 0.96–1.49, *p* = 0.95), nor were medians of OS (HR 1.16, 95% CI 0.88–1.53). Furthermore, the grade ≥3 treatment-related AEs were high and occurred in 41.8% vs. 13.9%. The study was stopped for futility [87].

Nivolumab was tested in a global phase II CheckMate 9KD trial, which enrolled 84 patients with chemotherapy-naïve mCRPC, ongoing androgen deprivation therapy and ≤2 prior novel hormonal therapies (NHTs). Participants received nivolumab (360 mg) and docetaxel (75 mg/m^2^) every 3 weeks with prednisone (5 mg) twice daily (≤10 cycles) and then nivolumab (480 mg) every 4 weeks (≤2 years). The confirmed ORR was 40.0%, and the confirmed PSA50-RR (prostate-specific antigen response rate; ≥50% decrease from baseline) was 46.9%. The median rPFS and OS were 9.0 and 18.2 months, respectively. Furthermore, without prior NHT, the ORR was 38.7% versus 42.9%, and the PSA50-RR was 39.6% versus 60.7% (PSA50-RR; ≥50% decrease from baseline). This trial proved that the combination of nivolumab plus docetaxel has clinical activity in patients with chemotherapy-naïve mCRPC. However, it only provides supportive data for the phase III double blind trial CheckMate 7DX, which is ongoing [88]. In mCRPC, the combination of nivolumab + ipilimumab in CheckMate 650 trial showed some activity in 90 patients with post-chemotherapy mCRPC, particularly those with a high tumor mutational burden (TMB), but early toxicity contributed to treatment discontinuations. The median overall survival was 19.0 and 15.2 months in the group with placebo. Grade 3–4 treatment-related adverse events occurred in 42–53% of patients, with four treatment-related deaths [89]. In additional results, nivolumab + ipilimumab was compared to monotherapy ipilimumab and to cabazitaxel; however, detailed evaluations of the characteristics of responders to immunotherapy doublet are warranted [90]. The pilot stage is also finished in a SOGUG randomized trial of ADT plus docetaxel +/− nivolumab or ipilimumab-nivolumab in high-volume metastatic, hormone-sensitive prostate cancer (hvHSPCa), but this trial needs to be expanded [91].

The last ICI tested on a population of patients with prostate cancer is atezolizumab. In the phase III study IMbassador250, atezolizumab in combination with enzalutamide was compared to monotherapy with enzalutamide in a cohort of men with mCRPC. The addition of atezolizumab did not improve OS in the whole population (stratified HR 1.12, 95% CI (0.91, 1.37), *p*  =  0.28) but did have an acceptable safety profile. However, longer progression-free survival was achieved in patients with high PD-L1 IC2/3, CD8 expression and established immune gene signatures. In conclusion, the investigators suggest that careful patient selection could identify subgroups of patients who may benefit from the addition of ICI [92]. In the randomized, open-label phase III trial CONTACT-02, the combination of atezolizumab with cabazitaxel was compared to NHT (enzalutamide/abiraterone) in mCRPC patients, who progressed on a prior NHT, had extrapelvic nodal or visceral metastasis and had a poor prognosis with limited, broadly available treatment options beyond chemotherapy. Median PFS was significantly longer in the ICI arm (6.3 months vs. 4.2 months; HR 0.65, 95% CI 0.50–0.84; *p* = 0.0007), including in subgroups with liver metastasis (6.0 months vs. 2.1 months; HR 0.47 [95% CI 0.30–0.74]) or prior docetaxel treatment for mCSPC (8.8 months vs. 4.1 months; HR 0.55 [0.32–0.96]). The ORR was also higher in the experimental arm (13.6 vs. 4.2%). OS data are now immature [93].

As we observe, apart from the fact that patients with a high TMB show a favorable response to ICI compared with taxanes alone in metastatic prostate cancer and that TMB over 10 mt/Mb is connected with significantly longer time to the next treatment and OS [94], ICI cannot be considered as a universal booster to standard therapy for the whole population. Only CONTACT-02 seems to be a positive trial; however, data are immature and need to be evaluated. The hypotheses why immunotherapy trials were not successful in prostate cancer are based on the natural course of that malignancy. The first is an unsuitable environment for tumor-infiltrating immune cells with anti-tumor activities, especially a smaller number of tumor-infiltrating CD8+ T cells and increased numbers of immunosuppressive cells, including tumor-associated macrophages, regulatory T cells and myeloid-delivered suppressor cells [95]. The second is the theory that cold tumors like prostate cancer, with fewer somatic mutations (a low TMB), have generally decreased responses to immunotherapy, in opposition to malignancies with a high TMB. As we mentioned, the high TMB tumors express more neoantigens, which lead to an increased chance of T-cell activation. In addition, the PD-L1 expression is low, while the level of PD-L2 expression is significantly higher in prostate cancer cells, which can be explained by relatively low levels of proinflammatory cytokines, secreted from CD8+ T cells [95].

## 7. Germ Cell Tumors–Dashed Hopes

Testicular cancer has one of the best cure rates (90%) and a >95% 5-year survival rate in systemic therapy for neoplasms. Despite the good chemotherapy response, some become chemo-resistant and progress through standard lines of therapy. Sometimes that phenomenon could be connected with a high TMB [3]. In the phase II KEYNOTE-158 study, previously pretreated patients with advanced solid tumors, including 12 with nonseminoma, who progressed after first-line therapy, were selected. They received pembrolizumab (200 mg) every 3 weeks. Patients with a high TMB (≥10 mutations per megabase) presented a 30% ORR, which led to selecting that group for potential benefit from ICI therapy [96]. The phase II KEYNOTE-158 study also focused on pembrolizumab in patients with previously treated, advanced noncolorectal MSI-H/dMMR cancer, including one patient with nonseminoma. Results in the entire population were encouraging: ORR was 34.3% (95% CI, 28.3% to 40.8%), median PFS was 4.1 months (95% CI, 2.4 to 4.9 months) and median OS was 23.5 months (95% CI, 13.5 months to not reached) [96]. The FDA granted accelerated approval to pembrolizumab for first tissue/site agnostic indication (including testicular cancer) [97] and subsequently for adults and children with TMB-H solid tumors [98]. The first single-arm phase II trial cohorted 12 patients with germ cell tumors (all patients had nonseminoma) and no curable options to treatment with pembrolizumab. Only two patients achieved radiographic stable disease, respectively, but with an increased level of AFP (alpha-fetoprotein). Pembrolizumab was well tolerated but does not appear to have had clinical activity in refractory germ cell tumors [99]. In the next phase II study, patients with multiple relapsed and/or refractory germ cell tumors were treated until progression or unacceptable toxicity. From a group of fifteen patients, all had disease progression. The twelve-week PFS was 0%, median PFS was 0.9 months (95% CI 0.5–1.9), and median OS was 2.7 months (95% CI 1.0–3.3). The study failed and suggested a lack of avelumab efficacy in unselected multiple relapsed/refractory germ cell tumors [100]. Avelumab was also investigated in Cohort A of the TROPHIMMUN Phase II Trial focused on gestational trophoblastic tumors. Patients who experienced disease progression after single-agent chemotherapy received avelumab until human chorionic gonadotropin (hCG) normalization, followed by three consolidation cycles. Eight out of fifteen patients (53.3%) had hCG normalization after a median of nine avelumab cycles; none subsequently relapsed. The response was not connected with a disease stage, FIGO (Federation of Gynecology and Obstetrics) score, or baseline hCG. Avelumab had a favorable safety profile [101]. An open-label randomized phase II study, APACHE, tested and compared durvalumab monotherapy and in combination with tremelimumab in patients with advanced germ cell tumors. The investigation cohorted 11 patients in arm A (monotherapy) and 11 in arm B (combination). The trial failed because 100% had progression in arm A and 82% in arm B until the control point [102].

The checkpoint inhibitors have not met the expectations of the clinicians. Only avelumab, and only in gestational trophoblastic tumors, has shown meaningful clinical benefit. A possible explanation for this observation lies in the unique microenvironment of germ cell tumors, which is characterized by a physiologically suppressed immunologic microenvironment. Although increased vascularization is probably the response to upregulation of PD-L1 in testicular tumors, the impaired blood–testis barrier has the capacity for an influx of cytotoxic immune cells into tumor tissue but without a chance of efflux, which leads to a reduction in effectiveness of ICI therapy [103]. The theory of cold tumors with a low TMB could be also applied to germ cell tumors [104].

## 8. Cost-Effectiveness of ICI Therapy in Urological Cancers

The implementation of ICI therapy in urological cancers was connected with some economic burden in multiple countries. Cost-effectiveness is varied between countries. The highest price of pembrolizumab in the second-line treatment of urothelial cancer was noticed in the United States and the lowest in Australia [105]. In some countries, the ICI therapy is considered a cost-saving option; an example is the doublet therapy of nivolumab plus ipilimumab in first-line advanced RCC treatment for patients with intermediate and poor risk profiles in Uruguay [106]. On the other hand, a study by Dharma Gupta and colleagues suggests that sunitinib has better cost-effectiveness than combination therapy in RCC, pembrolizumab/lenvatinib, and nivolumab/ipilimumab [107]. The dynamic development of new regimens in therapy for UC could change the pharmacoeconomy of some healthcare systems; however, the conclusions will be possible in the near future.

## 9. Future Perspectives

The future will probably bring a new era of bi-specific antibodies. Examples include anti-PD-1/anti-VEGF-2 [108,109] and anti-PD-1/CTLA-4 [110]. In addition, clinical trials with other agents including TKIs, anti-VEGF and chemotherapy will be most likely investigated, especially in subsequent lines of therapy. One of the most promising ways lies in combination with PARP inhibitors, especially in prostate cancer [85]. Further investigations will show new horizons and direction of immunotherapy in urological cancer.

## 10. Materials and Methods

The literature search was selected using the following electronic databases: PubMed, Scopus, and Web of Science. We included peer-reviewed articles published between 1992 and 2024, written in English, that addressed the use of checkpoint inhibitors in the treatment of urological cancers, their molecular structure and cellular pathways. Both clinical trials and high-quality narrative or systematic reviews were considered.

Studies were included if they met the following criteria: studies involving adult patients with urological cancers, articles discussing checkpoint inhibitors as monotherapy or in combination, reviews and clinical trials with clearly reported outcomes. Exclusion criteria were non-English publications, case reports, editorials, studies focusing on non-epithelial histologies or unrelated cancer types.

## 11. Conclusions

The introduction of checkpoint inhibitors has opened a new era for some urological cancers, such as renal cell cancer or urinary bladder cancer. In other clinical trials, the results were disappointing and did not change the current standard. The new form of therapy is evolving and shows new horizons in the treatment of urological cancers. In renal cell cancers, immunotherapy doublets or combinations with tyrosine kinase inhibitors are new ways to prolong overall survival. Consolidation with avelumab after response to platinum-based chemotherapy and combination with enfortumab vedotin is the new paradigm. In prostate cancer, despite attempts, immunotherapy has not become an optimal booster at this moment. In germ cell tumors, only avelumab in gestational trophoblastic tumors shows some activity; however, there is no therapy revolution. Future studies will probably extend the deployment of immunotherapy in urological cancer.

## Figures and Tables

**Figure 1 ijms-26-07194-f001:**
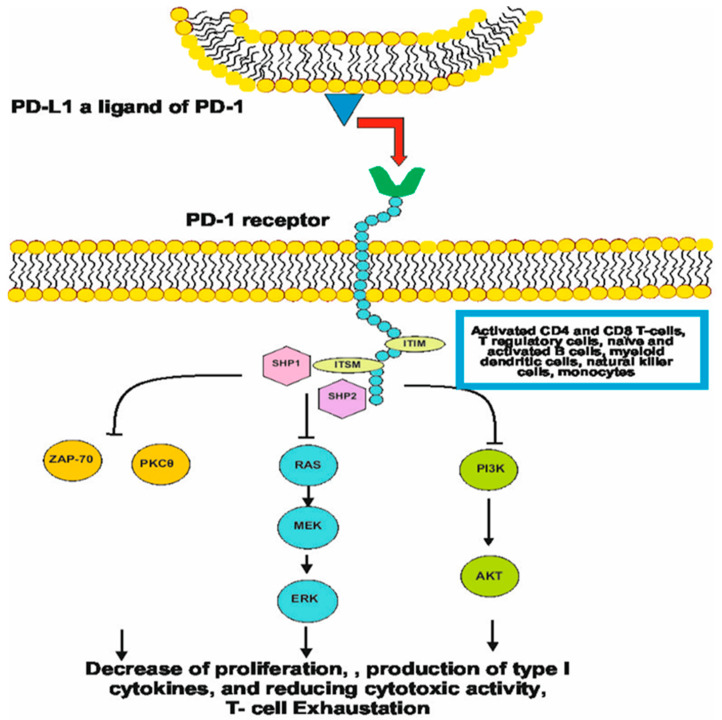
The process of ligation PD-1 to ligand.

**Table 1 ijms-26-07194-t001:** Positive phase III clinical trials focused on ICI in therapy for RCC.

Acronym	Line of Therapy	Agents	Authors
KEYNOTE-564	Adjuvant	Pembrolizumab	Tomczak P. et al., 2022 [59]
CheckMate 214	I	Ipilimumab + Nivolumab	Escudier et al., 2015 [46] Tannir et al., 2024 [47]
Chemate 9ER	I	Nivolumab + Cabozantinib	Choueiri et al., 2021 [51]
KEYNOTE-426	I	Pembrolizumab + Axitinib	Rini et al., 2019 [49]
CheckMate 025	II or subsequent	Nivolumab	Motzer et al., 2015 [44]
CLEAR	III	Pembrolizumab + Levantinib	Motzer et al., 2021 [54]

**Table 2 ijms-26-07194-t002:** Positive clinical trials focused on ICI in therapy for UCC.

Acronym	Line of Therapy	Agents	Authors
JAVELIN Bladder 100 (III phase)	After response to platinum-based chemotherapy	Avelumab	Powles et al., 2020 [75]
EV-302 (III phase)	I line	Pembrolizumab + enfortumab vedotin	Powles et al., 2024 [80]
KEYNOTE-052 (II phase)	I line (ineligible for platinum-based chemotherapy)	Pembrolizumab	Vuky J. et al., 2020 [65]
JAVELIN Solid Tumor (III phase)	II line (after progression for platinum-based therapy)	Avelumab	Patel et al., 2018 [69]
CheckMate 275	II line (after progression for platinum-based therapy)	Nivolumab	Sharma et al., 2017 [73]
CheckMate 901	I line with cisplatinum-gemcytabine	Nivolumab	van der Heijden et al., 2023 [79]
CheckMate 274	Adjuvant after radical surgery	Nivolumab	Bajorin et al. 2021 [43]

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
