# Peer review of "Immune Checkpoint Inhibitors (ICI) in Urological Cancers: A New Modern Era, but Not Generally Applied"

_ijms, 2025, doi:10.3390/ijms26157194_

Round 1

Reviewer 1 Report

Comments and Suggestions for Authors
  • Expand on Limitations and Challenges: While the review mentions that ICI are "not generally applied" or have had "disappointing" results in certain cancers (like prostate or germ cell tumors), expanding on the specific reasons for these limitations, such as resistance mechanisms or lack of predictive biomarkers, would be beneficial.

  • Discuss Biomarkers in More Detail: Although MSI-H/dMMR and TMB are mentioned for prostate cancer, a more comprehensive discussion on current and emerging biomarkers for ICI response across all urological cancers would enhance the review's value. This could include PD-L1 expression, tumor mutational burden (TMB), microsatellite instability (MSI), and other potential indicators.

  • Include Economic Considerations: A brief mention of the economic burden and cost-effectiveness of ICI therapies, especially combination regimens, could add another layer of practical relevance for clinicians and healthcare systems.

  • Consider Real-World Evidence: While clinical trials are the gold standard, briefly touching upon real-world evidence and its implications for ICI use in heterogeneous patient populations could offer a broader perspective.

Author Response

Dear Editor,

We would like to thank you for comprehensive reviev of our manuscript. After thorough analysis, we have implemented a corrections based on your comments. Bellow we want to present our adjusments:

  • Expand on Limitations and Challenges: While the review mentions that ICI are "not generally applied" or have had "disappointing" results in certain cancers (like prostate or germ cell tumors), expanding on the specific reasons for these limitations, such as resistance mechanisms or lack of predictive biomarkers, would be beneficial.

  • Response 1: We have added few sentences focused on limitations and resistance mechanisms in prostate and germ cell cancer with adequate citations
  • Discuss Biomarkers in More Detail: Although MSI-H/dMMR and TMB are mentioned for prostate cancer, a more comprehensive discussion on current and emerging biomarkers for ICI response across all urological cancers would enhance the review's value. This could include PD-L1 expression, tumor mutational burden (TMB), microsatellite instability (MSI), and other potential indicators.

  • Response 2:We have added a chapter about biomarkers (PD-L1 expression, TMB,,MSI)
  • Include Economic Considerations: A brief mention of the economic burden and cost-effectiveness of ICI therapies, especially combination regimens, could add another layer of practical relevance for clinicians and healthcare systems.

  • Response 3 : We added a short chapter focused on economical effectivness of ICI in urological cancer
  • Consider Real-World Evidence: While clinical trials are the gold standard, briefly touching upon real-world evidence and its implications for ICI use in heterogeneous patient populations could offer a broader perspective.

  • Respoce4: After revision, we have added some Real World evidence studies to our reviev

We hope our article is now suitable to your journal.

Yours sincerely,

Marcin Sokołowski

Reviewer 2 Report

Comments and Suggestions for Authors

The authors examined prominent ICI clinical trials for urological cancers and provided an in-depth list of trial out comes, but failed to give further discussion or more insight into the results other than which trials look promising and which ones did not. The overall English could use some improvement, and certain sentences are incomplete (e.g.line 140). Overall the manuscript is informative in its own way, but could use some significant improvement.

Specific comments follow:

  • The authors used the phrase systemic therapy a lot in the first half of the manuscript without providing a clear definition.
  • When first introducing PD-L1, the author failed to mention it is expressed by myeloid cells, which probably more relevant to ICI than expression on B cells.
  • When describing PD-1 structure the N-terminal extremity sequence that contains ITAM is in the middle of the protein, better description should be used here.
  • The authors mentioned PD-L1’s role in CRC without providing any citations.
  • The authors mentioned PD-L1 can also enhance T cell activation without providing any substantial discussion. This could be important for why certain therapies fail to improve outcomes in certain cancer types.
  • Overall, there is a lack of in-depth discussion on the outcomes of trials, why are some less promising than others , what factors could be associated with the outcomes, what can be improved etc.

Author Response

Dear Editor,
We would like to thank you for comprehensive reviev of our manuscript. After thorough analysis, we have implemented a corrections based on your comments. Bellow we want to present our adjusments:

Comments 1: The authors used the phrase systemic therapy a lot in the first half of the manuscript without providing a clear definition.

Response  1: We added definition in Introduction part

Comments 2:When first introducing PD-L1, the author failed to mention it is expressed by myeloid cells, which probably more relevant to ICI than expression on B cells.

Response 2; We correct that omission and added adequate citation

Comment3; When authors describing PD-1 structure the N-terminal extremity sequence that contains ITAM is in the middle of the protein, better description should be used here.

Comments 3: We have expanded a desceiption of N terminal extreminty in our reviev

Comments 4:The authors mentioned PD-L1’s role in CRC without providing any citations.

Response 4: We provide a new citation focused on role PD-L1 in CRC

Coments 4:The authors mentioned PD-L1 can also enhance T cell activation without providing any substantial discussion. This could be important for why certain therapies fail to improve outcomes in certain cancer types.

Response 4: We have expanded a desctripionod T cell activations

Comment 5:Overall, there is a lack of in-depth discussion on the outcomes of trials, why are some less promising than others , what factors could be associated with the outcomes, what can be improved etc.

Response 5: We inserted few sententences focused on factor which could be accociated with clinical

trial outcomes (resistance mechanisms, tumor micrioenvioent)

We hope our article is now suitable to your journal.
Yours sincerely,
Marcin Sokołowski

Round 2

Reviewer 2 Report

Comments and Suggestions for Authors

The authors have addressed all my concerns.